# Characterization of locomotor phenotypes in zebrafish larvae requires testing under both light and dark conditions

**Melek Umay Tuz-Sasik**[ORCID]*, **Henrik Boije, Remy Manuel**\*

Department of Immunology, Genetics and Pathology, Cell and Neurobiology, Uppsala University, Uppsala, Sweden

\* umaytuz@gmail.com (MUTS); remym@me.com (RM)

**Data Availability Statement:** All relevant data are within the paper and its Supporting Information files.

**Funding:** RM: Olle Engkvists Stiftelse 204-0243 (https://engkviststiftelserna.se) MUT-S: Brain

## Abstract

Despite growing knowledge, much remains unknown regarding how signaling within neural networks translate into specific behaviors. To pursue this quest, we need better understanding of the behavioral output under different experimental conditions. Zebrafish is a key model to study the relationship between network and behavior and illumination is a factor known to influence behavioral output. By only assessing behavior under dark or light conditions, one might miss behavioral phenotypes exclusive to the neglected illumination setting. Here, we identified locomotor behavior, using different rearing regimes and experimental illumination settings, to showcase the need to assess behavior under both light and dark conditions. Characterization of free-swimming zebrafish larvae, housed under continuous darkness or a day/night cycle, did not reveal behavioral differences; larvae were most active during light conditions. However, larvae housed under a day/night cycle moved a shorter distance, had lower maximum velocity and maximum acceleration during the startle response under light conditions. Next, we explored if we could assess behavior under both dark and light conditions by presenting these conditions in sequence, using the same batch of larvae. Our experiments yielded similar results as observed for naïve larvae: higher activity during light conditions, regardless of order of illumination (i.e. dark-light or light-dark). Finally, we conducted these sequenced illumination conditions in an experimental setting by characterizing behavioral phenotypes in larvae following neuromast ablation. Depending on the illumination during testing, the behavioral phenotype following ablation was characterized differently. In addition, the results indicate that the order in which the light and dark conditions are presented has to be considered, as habituation may occur. Our study adds to existing literature on illumination-related differences in zebrafish behavior and emphasize the need to explore behavioral phenotypes under both light and dark condition to maximize our understanding of how experimental permutations affect behavior.

foundation FO2020-0129 (https://www.hjarnfonden.se) HB: Swedish research council 2020-03365 (https://vr.se) HB: Ragnar Söderberg foundation 1235/17 (https://ragnar.soderbergs.org) HB: Brain foundation FO2020-0129 (https://www.hjarnfonden.se)

**Competing interests:** The authors have declared that no competing interests exist.

## Introduction

Zebrafish is a powerful model organism to test behavioral outcomes of subtle changes in neuronal pathways (see review, [1]). For example, neurotoxic compounds have been widely tested in zebrafish larvae in terms of not only survival and phenotypic alterations, but also behavioral outcomes [2–4]. Several studies indicate that behavioral parameters of zebrafish larvae change depending on the time of day, animal age, previous rearing conditions, and well size during behavioral experiments [4–6]. Illumination differences, such as intensity (dark *vs* light), duration (long *vs* short exposure) and dark/light cycle (continuous *vs* alternating) also affect the behavior of zebrafish larvae [5, 7, 8]. Generally, the locomotor activity is higher under light compared to dark conditions [5, 9]. However, there are several studies, utilizing an alternating light/dark cycle, which reported higher locomotor activity during the dark period [5, 8, 10, 11]. Due to these illumination-related differences in behavioral output, the possibility exists that locomotor phenotypes are not detected, simply because one illumination condition was neglected.

Here, we highlight the importance of assessing zebrafish larvae behavior under both light and dark conditions and provide an experimental pipeline to do so using the same batch of animals. Zebrafish larvae were housed under either continuous darkness or a 14/10-hour day/night cycle and free-swimming and startle response were assessed both during illumination and in darkness. Both housing conditions resulted in higher activity of larvae under light test conditions, compared to those under dark test conditions. Startle response was only significantly lower under light conditions in larvae housed under a day/night cycle. Next, we assessed if these differences remained when larvae behavior was assessed in a paradigm where light and dark conditions were presented in sequence using the same batch of fish. Here too, we observed that larvae tested under light conditions had the higher activity during free-swimming, and a lower activity during the startle response. However, we also observed that the activity level in the second part of the paradigm was different to the first part, when comparing the same illumination condition. This indicates there may be a form of habituation at play.

Lastly, larvae were exposed to neomycin to ablate the neuromasts, which is known to induce a behavioral phenotype [12]. Our data revealed significant differences in free-swimming parameters between control and treated larvae under dark condition, if larvae were naïve. Thus, we observed no differences in under dark conditions, if this was presented at the end of the sequential paradigm, causing our phenotype to disappear. We observed no significant differences under light conditions, regardless of the order of illumination. Interestingly, during startle responses, significant differences were more pronounced under light conditions, in both naïve and pre-exposed larvae. The presence of visual input likely compensates for the loss of neuromasts during free-swimming, but not startle behavior. Thus, by only analyzing free-swimming behavior, and doing this under only light conditions, would have resulted in inadequate characterization of the phenotype. These results add to the existing literature regarding altered behavior during different lighting conditions and showcase the importance of assessing phenotypes under both. Based on our data, this is best done with naïve larvae, as we observed habituation effects during our sequential illumination paradigm, which interfered with the characterization of the phenotype.

## Materials and methods

### Animals and housing

Adult AB zebrafish of Tg(HGn36A) were housed at the Genome Engineering Zebrafish National Facility (SciLifeLab, Uppsala, Sweden) under standard conditions of 14/10 hours

day/night cycles at 28°C. Embryos for behavioral experiments were obtained from group breeding (in-cross) and kept at 28°C (under constant darkness) until 6 days post fertilization (dpf) unless otherwise stated. The larvae were not fed before or during the experiment. Appropriate ethical approvals were obtained from a local ethical board in Uppsala (permit number: 14088/2019)

## Housing conditions (day/night)

When testing lighting conditions during housing prior to behavioral experiments, embryos were kept under either constant darkness or under a 14/10 hours day/night cycle at 25°C, until 7 dpf. Both groups of larvae were kept in the fish room at 25°C, one group covered, to benefit from the lighting routine. To compensate differences of development at the lower temperature, animals were tested at 7 dpf, which approximately corresponds to the developmental stage of 6 dpf at 28°C based on the linear function as previously established [13].

## Ablation of neuromasts through neomycin treatment

Neomycin sulfate treatment is frequently used to ablate the superficial neuromast cells [14]. A 100 mM stock solution of neomycin sulfate (PHR1491, Sigma Aldrich, St. Louis, MO, USA) was prepared in $dH_2O$. At 6 dpf, larvae were exposed to 500 μM final concentration of neomycin sulfate, in 5 ml E3 medium including methylene blue, for 1 hour. Then, the neomycin sulfate solution was replaced with fresh E3 medium with methylene blue for a 3-hour recovery prior to behavioral testing. All steps were conducted under constant darkness at 28°C. Following the behavioral analysis, larvae were exposed to fluorescent dye DASPEI (D0815, Sigma Aldrich, St. Louis, MO, USA), which labels neuromast cells [14], to verify successful ablation.

## Assessment of free-swimming and startle response

Behavioral tests were performed using a DanioVision behavioral chamber (Noldus Information Technology, Wageningen, the Netherlands) under darkness (infrared light) or light conditions (100% intensity ~ 4400 lux) at 25 frames per second. Experimental conditions were similar to those described previously [15], with some minor modifications. Briefly, individual larvae were distributed (in a checker-board fashion for experiments containing control and experimental groups) to wells of a 48-well plate containing 1 ml of embryo water. Once placed in the DanioVision, larvae acclimatized at 28°C for 50 min in light or dark conditions, depending on the experiment. Each trial included free-swimming for 10 min followed by a sequence of three taps, at 6-minute intervals, to elicit startle responses under the same illumination as the acclimation period. Tapping was done mechanically via a solenoid inserted at the bottom of the base of the DanioVision chamber [16]. Stimulus intensity was at the level of 8 (most intense) for all experiments. For the sequential illumination test, larvae were given a 20-min acclimation period prior to assessing behavior under the illumination condition opposite to initial one. All behavioral experiments were performed between 1 and 6 pm. Two independent trials were performed for each condition on different days from different parent fish. Also, in order to avoid time of day effect, the order of illumination was alternated between independent trials, i.e., dark conditions at 1 pm at the first trial and light conditions at 1 pm for trial number two.

Larvae were tracked by Noldus EthoVision XT (version 13, Noldus Information Technology, Wageningen, the Netherlands) using dynamic subtraction as the detection method. To remove system noise, all data were filtered with a threshold of 0.200 mm (minimum distance moved). In order to exclude faulty detections, manual filtering was performed using EthoVision's "integrated visualization" feature. During the last 10 minutes of free-swimming the

following parameters were analyzed: total displacement, maximum velocity, maximum acceleration, maximum deceleration, cumulative time accelerating, number of accelerations, number of movements, and cumulative movement duration. For startle responses, the following variables were analyzed over a period of 280 ms after each tap: total displacement, maximum velocity, maximum acceleration, cumulative time accelerating, and cumulative velocity. An average of three taps was used for analysis of each larvae.

## Statistical analyses

Statistical analyses and plotting of data were performed using GraphPad Prism 9 for Windows (GraphPad Software, La Jolla, CA, USA). First, the outliers were identified with Grubbs'-method ($\alpha = 0.05$) and data were tested for normal distribution with Kolmogorov-Smirnov test. When assumptions were fulfilled, unpaired *t*-tests (two-tailed) were performed. In the case of comparing data coming from the same larvae, paired *t*-tests (or non-parametric Wilcoxon matched-pairs signed rank tests) were used. In the cases where the assumptions of normality were not fulfilled, a non-parametric Mann-Whitney-U test was applied. All individual values are presented in scatterplots. For parametric tests mean values of groups are presented, whereas median values are shown for non-parametric tests. The level of significance was set at $p \leq 0.05$ (two-tailed) (*$p \leq 0.05$, **$p \leq 0.01$, ***$p \leq 0.001$, ****$p \leq 0.0001$). When multiple comparisons were done, the level of significance was corrected manually. Statistics can be found in S1 Table.

## Results

### Zebrafish larvae exhibit higher swim activity under light conditions

In order to establish the baseline behavior in light or dark, larvae were raised under constant darkness until the behavioral trial (Fig 1A). During a 10-min free-swimming episode, larvae under light conditions were significantly more active than those in darkness, as indicated by a greater total distance travelled (p<0.0001), a higher max velocity (p = 0.0007) and max acceleration (p = 0.0051; Fig 1B), as well as greater output in related parameters such as: maximum deceleration, accelerating frequency, time accelerating, moving frequency, and time moving (Fig 1C). However, in the case of the startle responses none of the parameters were significantly different under light conditions compared to those that took place in darkness (Fig 1D and 1E). Overall, our data show clear differences between light and dark conditions during assessment of free-swimming in larvae zebrafish.

Previous studies indicate that housing conditions, such as illumination [17], may alter behavior in experiments. To identify if lighting conditions during housing changes the behavioral phenotypes under light and dark test conditions, we raised larvae under constant darkness or a 14/10-hour day/night cycle (Fig 1A). Our data reveal that illumination-related differences during behavioral testing of free-swimming larvae remained and that characterization of the behavioral phenotype was the same following both housing conditions (Fig 1B and 1C). However, we did characterize the phenotype differently for the startle response, depending on our housing condition. Larvae housed under continuous darkness had similar responses under light and dark conditions while larvae housed under a day/night cycle showed a significantly lower total distance moved (p = 0.0012), maximum velocity (p = 0.0018), maximum acceleration (p = 0.0018), and cumulative velocity (p = 0.0012), under light conditions (Fig 1D and 1E). Combined, these data show that free-swimming larvae have higher activity under light compared to dark test conditions, but that the startle response to the tapping stimuli is weaker under such lighting.

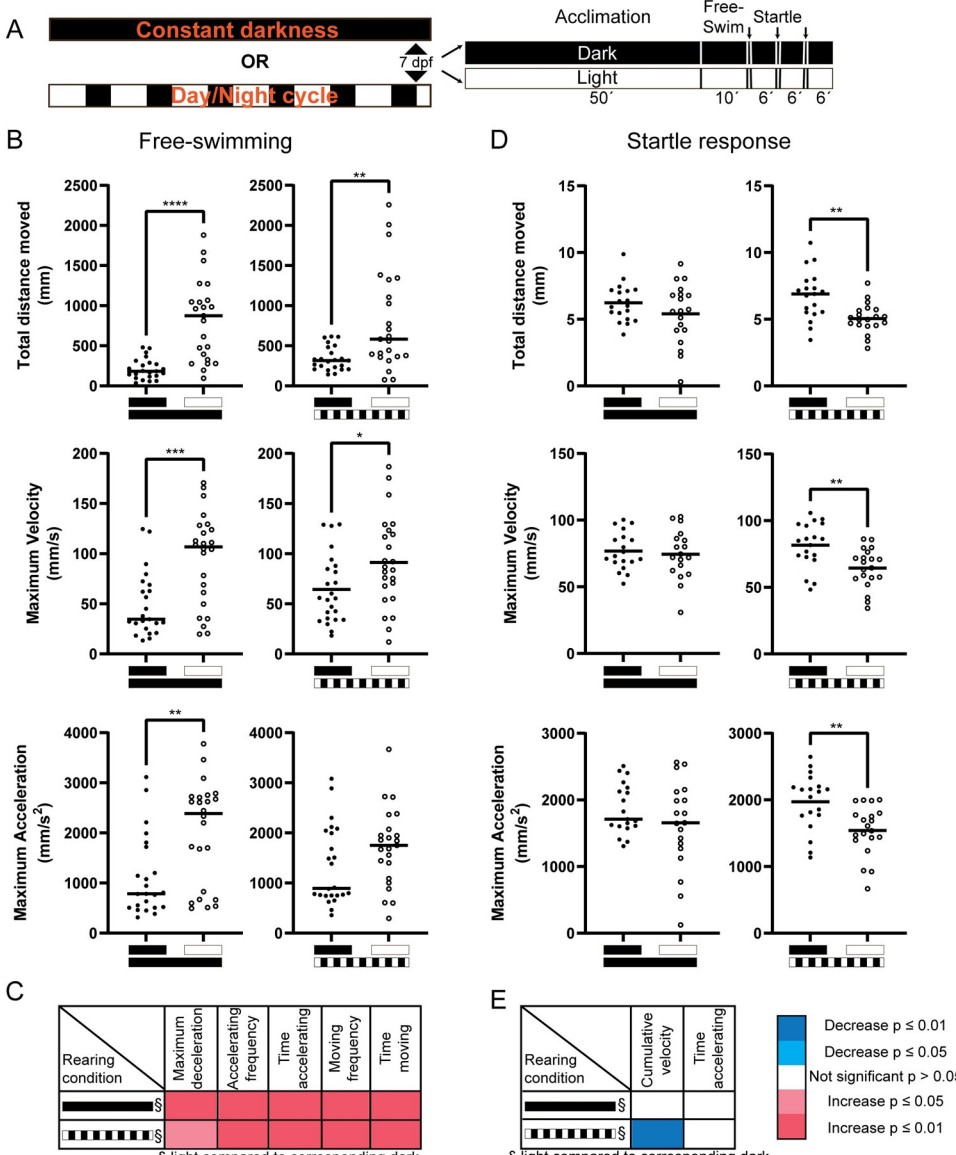

**Fig 1. Larvae exhibit higher activity under light conditions.** (A) Schematic of the experimental procedure. Embryos were kept under constant darkness (solid black bar) or a 14/10 h day/night cycle (striped bar) until the behavioral experiments. Each behavioral section had two components; free-swimming for 10 min under light (open circle; white) or dark (solid circle; black) conditions after a 50 min acclimation period under same illumination condition, and a set of startle responses (280 ms) where a sequence of three taps was executed with 6-minute interval; arrows designate timing of taps. (B, C) Both larvae housed under continuous darkness or a day/night cycle showed significantly more activity during free-swimming when tested under light conditions, compared to dark test conditions. (D, E) For larvae housed under continuous darkness we observed no differences in startle behavior, but larvae housed under a day/night cycle showed significantly lower activity during startle behavior when tested under light conditions, compared to dark test conditions. Free-swimming: n = 23–24 per group, startle response: n = 18–21 per group. Data is presented in scatterplots showing individual values and group mean or median. Significance: $^*p \leq 0.05$, $^{**}p \leq 0.01$, $^{***}p \leq 0.001$, $^{****}p \leq 0.0001$.

## Sequential exposure to light and dark reveals illumination-related behavioral phenotypes similar to that of naïve larvae

As behavioral assessment under both light and dark conditions is desired, we explored if both could be analyzed using the same batch of larvae, both to save time and reduce the number of animals needed. In order to examine this, we extended the behavioral paradigm to repeat itself with the opposite illumination condition and by reversing the order thereof (Fig 2A).

Regardless of larvae being exposed to the light or dark condition first, significantly increased activity was observed during free-swimming behavior during the light conditions (Fig 2B and 2C). Merging of the two data-sets from the same illumination, but different order, did not change the conclusions drawn from our results (S1A and S1B Fig). This suggest that merging can be performed to increase the number of larvae used during analysis, while still being able to alternate the order by which each illumination is presented between experimental runs. There were, however, differences observed during startle responses, where introducing the light conditions first, followed by darkness, appeared to cause a reverse in the observed behavior. Activity was lower during in darkness, opposed to it being higher in the light conditions when larvae were exposed to the darkness first (Fig 2D and 2E). Merging of the data on the startle response removed most of the significant differences seen (S1C and S1D Fig). Thus, free-swimming data appear consistent between our first and second experiments, while data regarding startle responses seems less consistent between experiments (Figs 1 and 2).

## Ablation of neuromasts resulted in a behavioral phenotype under dark, but not light, conditions

To test our ideas in practice we applied our sequential paradigm to identify behavioral phenotypes in zebrafish where neuromasts were ablated by application of neomycin (Fig 3A). Previous studies have demonstrated that neomycin treatment significantly change behavioral outcomes, both during free-swimming and startle responses [12, 14, 18]. To validate our paradigm, we compared behavioral data from control and treated larvae assessed under light and dark condition from naïve larvae as well as that of larvae from our sequential paradigm.

Our free-swimming data revealed that naïve neomycin treated larvae under dark conditions were less active than control larvae; indicated by significant differences across all parameters assessed (Fig 3B and 3F). No differences were found under light conditions (Fig 3C and 3F). When we then compare the data from the sequential light and dark conditions, we found that treated larvae assessed in darkness following a period of light did not show significant differences (Fig 3E and 3F). Under light conditions following a test period under dark conditions showed no significant differences (Fig 3D and 3F).

For naïve larvae under dark conditions, the analysis of startle behavior only revealed a significantly lower maximum velocity (p = 0.0193) and maximum acceleration (p = 0.0036) for treated larvae (Fig 4A). In contrast, the startle response under light conditions showed significantly decreased activity in treated larvae across all parameters assessed (Fig 4B and 4E). When we then compare the data from the sequential light and dark conditions, we found that treated larvae assessed under dark conditions had significantly lower values for all parameters analyzed (Fig 4D and 4E), except for the time accelerating (Fig 4E), which was not significant. Under light conditions all parameters analyzed (Fig 4C and 4E) were lower for treated larvae. Overall, the data show a strong reduction in activity of neomycin treated larvae, both for free-swimming and startle parameters. However, the phenotype found during free-swimming was lost in our sequential illumination paradigm, indicating the habituation affect observed previously, interferes with proper characterization of phenotypes.

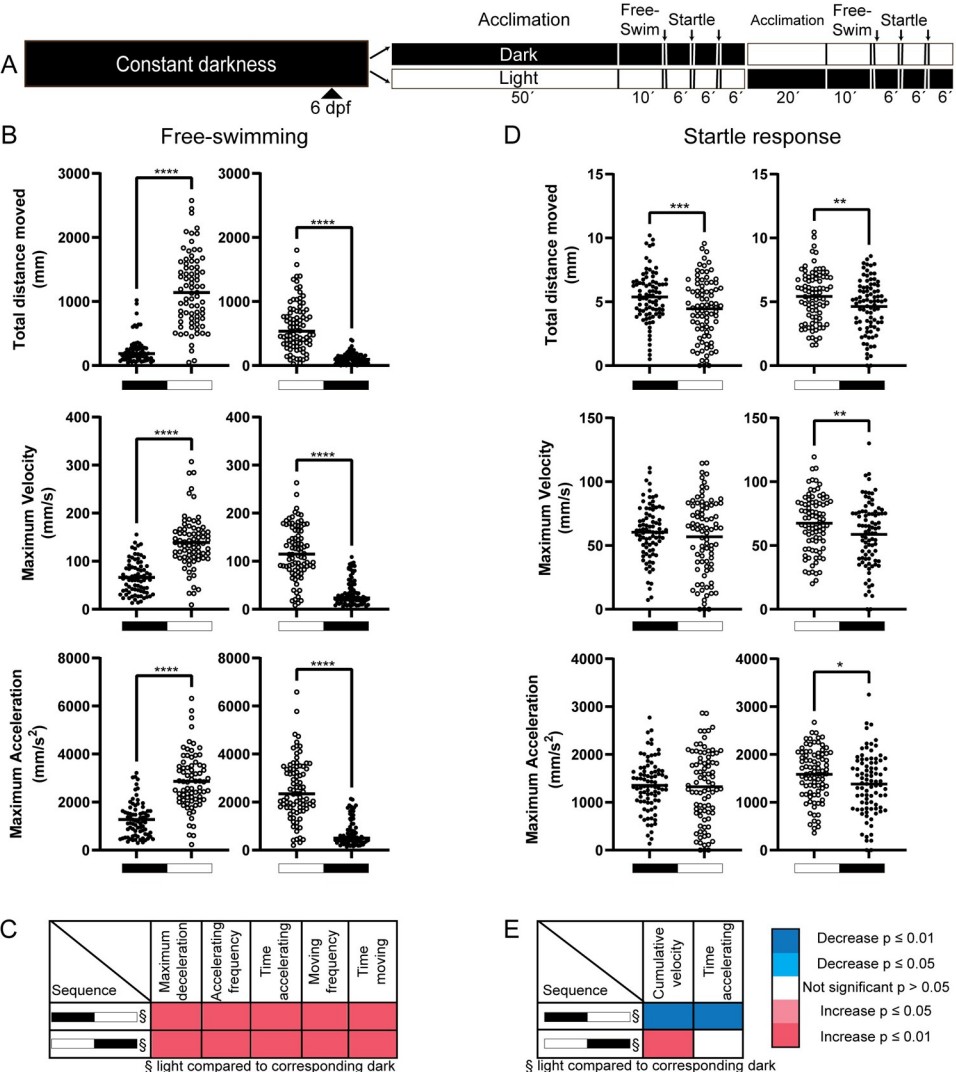

**Fig 2. Regardless of order, under light condition same larvae had higher free-swim activity than under darkness.**
(A) Schematic of the experimental procedure. Embryos were kept under constant darkness (solid black bar) until the behavioral experiments. Each behavioral section had two components; free-swimming for 10 min under light (open circles; white) or dark (solid circles; black) conditions after a 50 min acclimation period under same illumination condition, and a set of startle responses (280 ms) where a sequence of three taps was executed with 6-minute interval; arrows designate timing of taps. For the sequential illumination test, larvae were given a 20-min acclimation period prior to assessing behavior under the illumination condition opposite to initial one. (B, C) Regardless of the order of illumination, larvae showed significantly more activity during free-swimming when tested under light conditions, compared to dark test conditions. (D, E) In terms of startle response behavior, almost all parameters decreased in second condition (regardless of illumination) representing potential habituation effect. Free-swimming: n = 76–86 per group, startle response: n = 85–87 per group. Data is presented in scatterplots showing individual values and group mean or median. Analysis was performed with paired t-tests. Significance: $^*p \leq 0.05$, $^{**}p \leq 0.01$, $^{***}p \leq 0.001$, $^{****}p \leq 0.0001$.

## Discussion

### Free-swimming behavior under light and dark conditions

In the present study, we show that free-swimming larvae were more active under light conditions, which is in line with previous studies [5, 9, 19]. Zebrafish larvae are known to be phototactic, meaning that they prefer a lighter section of the tank compared to a darker section [20–

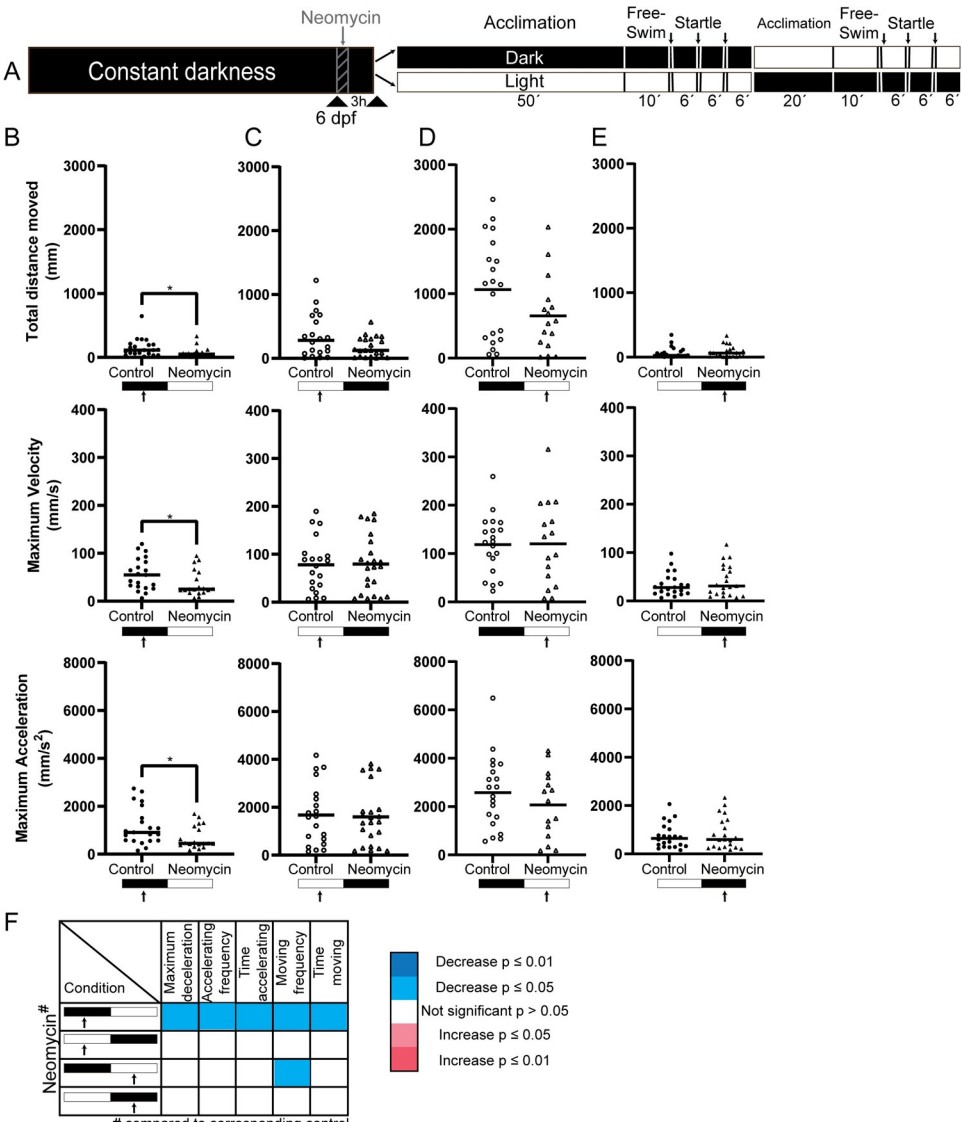

**Fig 3. Neuromast ablated larvae exhibited free-swimming phenotype under dark condition.** (A) Schematic of the experimental procedure. After 1 h of neomycin treatment embryos were kept in fresh embryo medium for 3 h to recover. Each behavioral section had two components; free-swimming for 10 min under light (open circles/triangles; white) or dark (solid circles/triangles; black) conditions after a 50 min acclimation period under same illumination condition, and a set of startle responses (280 ms) where a sequence of three taps was executed with 6-minute interval; arrows designate timing of taps. For the sequential illumination test, larvae were given a 20-min acclimation period prior to assessing behavior under the illumination condition opposite to initial one. (B) Neomycin treated larvae (triangles) had significantly less locomotor activity under initial darkness. (C, D) However, under light conditions both control and neomycin treated larvae can reach comparable maximum velocity and maximum acceleration regardless of order of light condition. (E) Larvae tested under dark condition followed by light did not show free-swim phenotype after neomycin treatment. (F) Heatmap representation of other analyzed parameters indicated only under first darkness, neomycin phenotype can be detected. n = 15–23 per group. Data is presented in scatterplots showing individual values and group mean or median. Significance: $^*p \leq 0.05$, $^{**}p \leq 0.01$, $^{***}p \leq 0.001$, $^{****}p \leq 0.0001$.

23] and have a tendency to swim towards light [24]. In addition, it was shown they lose visual responsiveness in darkness [19], which may lead to more exploratory behavior under brighter conditions.

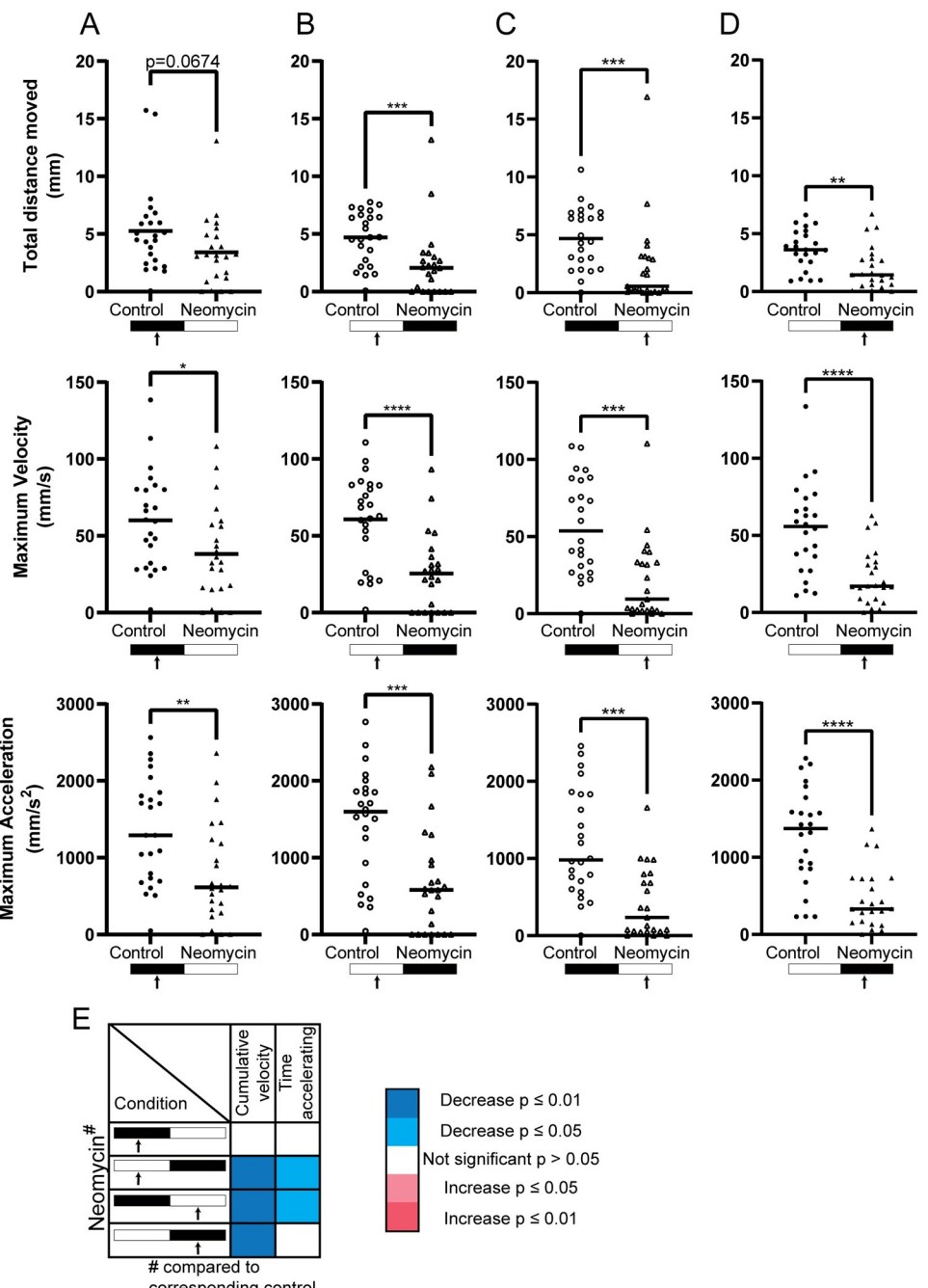

**Fig 4. Neuromast ablated larvae exhibited less startle response compared to control group.** (A, B, C, D) Neomycin treated larvae (triangles) had less response to tapping stimulus during 280 ms under both light (open triangles; white) or dark (solid triangles; black) regardless of order. (E) Heatmap representation of other analyzed parameters indicated the differences between control and treated group were evident under light. n = 23–24 per group. Data is presented in scatterplots showing individual values and group mean or median. Significance: *p ≤ 0.05, **p ≤ 0.01, ***p ≤ 0.001, ****p ≤ 0.0001.

Housing conditions, prior to behavioral experiments, have significant effects on behavioral outcomes [4, 17]. It was demonstrated that zebrafish larvae raised in constant darkness were less active than those housed under a day/night cycle [17]. In the same study, lighting

conditions during housing interacted with the behavioral characterization following bisphenol A (BPA) treatment [17]. It is common practice to house zebrafish embryos/larvae under constant darkness. To explore if housing larvae under a day/night cycle significantly altered the illumination-related behavioral phenotypes, we compared zebrafish larvae housed under constant darkness to those raised under a day/night cycle. Analysis of free-swimming behavior revealed similar phenotypes under light and dark test conditions following both housing conditions (Fig 1B). However, phenotypes appeared stronger following housing under constant darkness, signified by the smaller statistical p-values. Overall, our data suggest that introducing a day/night cycle does not remove differences in activity of larvae during light and dark test conditions.

### Free-swimming behavior under sequential dark and light conditions

Due to the differences in behavior observed for the different lighting conditions, and the lack of a clear understanding what underlies these, we suggest to assess behavior under both illumination conditions when screening for phenotypes. To reduce the time required and minimize the number of larvae used, we assessed if these two assays could be done in sequence with the same animals. In addition, we tested if the order of illumination affected the outcome of the phenotypes.

Regardless of the order in which larvae experienced the illumination conditions, activity was always higher under light condition, in agreement with our initial observation (Fig 2B). However, we observed an overall reduction in activity of groups that underwent the light-dark order of illumination, compared to those that underwent a dark-light order. In effect, the activity under light was greater when light was presented last, compared to when light was presented first. In contrast, activity under dark conditions was higher when presented first, compared to presented last (S2B and S2C Fig). While we currently have no explanation for these observations, habituation/acclimation to a novel environment [3, 25] or even the startle trigger [26] may play a role. Habituation is a non-associative form of learning defined as a decrease in behavioral response after repeated stimuli [27]. Since performing behavioral responses to riskless stimuli has an energy expenditure and increases motor fatigue, habituation is an essential mechanism to conserve energy and increase survival [28]. One could argue that habituation is stronger under light conditions, where visual input and increased exploratory behavior add to the exploration of a new environment. In contrast, increased anxiety caused by aversive light conditions may reduce habituation. Regardless, introducing larvae to different light conditions may interfere with their acclimation/habituation process, which in turn can influence their behavior during the second part of the paradigm. Alternatively, our experimental design may have introduced a confounding effect. For the first section of the paradigm, larvae were given 50 minutes of acclimation to the illumination used, whereas in the second part this was only 20 minutes. It is thus possible that, following the onset of light, larvae are more active after 20–30 minutes (measured in the second part of the paradigm) than they are after 50–60 minutes (measure in the first part of the paradigm). Similarly, larvae that experienced the dark condition following a period of light, may have already been exhausted, and now use the dark period to recover from the more active and possibly aversive light period. Keeping the acclimation period constant for both parts of the paradigm may prevent this confounding effect. Unfortunately, we did not record larvae during their habituation period, and we can thus only speculate on this.

Regardless, these observed differences are to be considered and likely interfere with proper characterization of phenotypes, as we showed following neuromast ablation (discussed later).

## The startle response under dark and light conditions

Our data indicate a stronger startle response to the tapping stimuli in darkness than during light conditions. However, we did observe that these illumination-related differences were not as consistent as for the free-swimming data. For instance, we found significant difference for some parameters in one experiment (strongest in larvae housed under a day/night cycle), but only trends at best for others (mostly when housed under constant darkness). In addition, when larvae underwent the light-dark paradigm (Fig 2D), the startle response yielded opposite results (i.e. light conditions had the greater response), which may be caused by habituation/acclimation or differences in acclimation period during this paradigm, as we discussed previously for free-swimming behavior.

The reason for a stronger startle response in darkness remains unknown. The higher activity prior to the startle response for larvae under light conditions may play a role. Larvae under light conditions have been increasingly active for 60 minutes prior to the startle trigger, which may have caused some form of exhaustion or motor fatigue, thereby reducing their overall ability to respond during a startle. Alternatively, it has been reported that light conditions may cause stress and anxiety-like behavior in larvae housed under constant darkness [5]. A greater state of stress may have reduced the response to the startle trigger [29], leading to a lower behavioral output of parameters measured. In addition, studies have shown that cortisol levels can either increase or decrease following acute exposure to light, depending on intensity and time of day [30], and that differences in plasma cortisol levels correlate to differences in behavioral and startle responses following a stressor [31–33].

However, these differences may simply relate to the ability of larvae to see their surroundings. In darkness, larvae receive no visual input [19] and have therefore less information regarding their environment. This lack of visual information may increase their startle behavior to a trigger, because they cannot see that there is no immediate threat, but must rely on senses picking up the vibrations caused by the tap. Regardless, these results stress the need to thoroughly plan behavioral experiments to maximize the output while minimizing habituation effects. In addition, measuring physiological parameters, such as whole-body cortisol, following behavioral studies should be considered.

## Free-swimming and startle behavior following neuromast ablation

To validate our approach in an experimental setting, we therefore assessed behavioral phenotypes in larvae where neuromasts had been ablated. Neomycin treatment ablates the superficial neuromast cells and causes behavioral alteration in zebrafish larvae [12, 14, 18]. In line with previous work [12], our naïve neuromast ablated larvae (i.e. those exposed to dark or light conditions in the first part of the paradigm), displayed a significant reduction in activity under dark conditions, but had no observable change in activity during light conditions (Fig 3B and 3C). Previously [12] it has been shown that neomycin damages muscle tissue, and it has been suggested that the decrease in free-swimming activity following neomycin treatment could be attributed to this, rather than the ablation of the neuromast. Note, that these conclusions were drawn following testing under dark conditions only. When we now look to the data we acquired under light conditions, we find that neomycin treated larvae performed similar to control animals (i.e. no significant differences between parameters analyzed). We can now argue that muscle damage is not likely the cause for the reduced free-swimming activity under dark conditions. Instead, we find it more likely that a lack of visual input under dark conditions [19], increased the reliance on the lateral line for adequate locomotion [34].

Next, we assessed the data from our sequential illumination paradigm to see if we would end up with the same phenotype. First, we characterized the phenotypes from the dark-light

illumination paradigm. Comparison between the groups under the light condition at the end of the paradigm suggested there is no effect of neomycin on free-swim behavior (Fig 3D), which is in line with our observations for naïve larvae. However, when we analyzed the data from the light-dark illumination paradigm, we no longer found significant differences under dark conditions (Fig 3E and 3F). We reason that the low activity of larvae under dark conditions in this illumination paradigm (as discussed previously) made a further reduction thereof following neuromast ablation, undetectable. We did attempt to merge the data as we did previously (S1 Fig), to see if perhaps significance would be restored by the greater number of animals (S3B and S3C Fig), but found this to be not the case.

Where we observed illumination-related effects on neomycin-induced behavioral phenotypes for free-swimming larvae, the outcome of the startle response was consistent across the illumination conditions (Fig 4). Moreover, the order in which illumination was provided (dark-light or light-dark) did not alter our identification of the phenotype. Here, sensory hair cell ablation caused a strong decrease in several parameters during startle responses, which is in line with previous observations [12].

In contrast to the free-swimming, where we believe the lack of feedback from functional neuromasts can largely be compensated by the visual system [34], the startle response is affected differently. The lateral line is able to directly activate the Mauthner cell [35], which is a key activator-cell of escape behavior in zebrafish [36]. Ablation of the neuromasts directly removes this pathway. In our study we used tapping on the dish as an escape trigger, which causes both an auditory signal and vibrations in the dish. These vibrations, however, can no longer be detected via the lateral line by neuromast-ablated larvae, which means this cue does not trigger the startle response. The use of a trigger that does not induce sound or vibrations, such as an electric shock or light-flash, may yield other results and is worth investigating in future studies.

## Concluding remarks

Although our study was performed in larvae, light and dark preferences also exists in adult zebrafish [37, 38]. Therefore, it is likely that effects of illumination during test conditions on behavioral phenotypes exist throughout the life of zebrafish. In addition, our findings can likely be extrapolated to other fish species or even mammalian models.

Proper behavioral analysis is crucial in order to characterize the phenotypes associated with mutations, morpholino injections, or experimental housing conditions. Here we highlight that behavioral characterization is best done under both light and dark test conditions. Not only will this increase the chance of finding behavioral phenotypes, as exemplified by a lack of difference under light conditions following neuromast ablation, but it also provides an additional stepping stone towards identifying the underlying mechanisms. Assessing the same batch of larvae in concession under both light and dark test conditions introduces confounding effects that can influence proper interpretation of the results. We therefore suggest assessing behavior of naïve animals under both dark and light conditions to properly identify and characterize behavioral phenotypes.

## Supporting information

**S1 Fig. Supplementary to Fig 2.** Merging of data from the same illumination following sequential testing of light and dark. (A) Experimental scheme of sequential application of illumination, light (open circles; white) and dark (solid circles; black). (B, C) Under light condition, larvae performed higher activity in free-swimming episode. (D, E) The differences were disappeared in terms of startle response. Free-swimming: n = 162–163 per group, startle

response: n = 172 per group. Data is presented in scatterplots showing individual values and group mean or median. Analysis was performed with paired t-tests. Significance: *p ≤ 0.05, **p ≤ 0.01, ***p ≤ 0.001, ****p ≤ 0.0001.
(TIF)

**S2 Fig. Supplementary to Fig 2.** Comparison of data from the same illumination following sequential testing of light and dark. (A) Experimental scheme of sequential application of illumination, light (open circles; white) and dark (solid circles; black). (B, C) In terms of free-swimming, second dark and second light condition had different strength compared to first order same illumination. (D, E) Regardless of illumination, second condition caused a decrease in startle response parameters. Free-swimming: n = 76–86 per group, startle response: n = 85–87 per group. Data is presented in scatterplots showing individual values and group mean or median. Analysis was performed with paired t-tests. Significance: *p ≤ 0.05, **p ≤ 0.01, ***p ≤ 0.001, ****p ≤ 0.0001.
(TIF)

**S3 Fig. Supplementary to Fig 3.** Merging of data from the same treatment (control vs neomycin) under similar illumination following sequential testing of light and dark. (A) Experimental scheme of sequential application of illumination, light (open circles/triangles; white) and dark (solid circles/triangles; black), after neomycin treatment (triangles). (B, C) All significances were disappeared under dark condition. (D, E) The significant differences were remained under both dark and light condition in case of startle response. Free-swimming: n = 35–39 per group, startle response: n = 47–48 per group. Data is presented in scatterplots showing individual values and group mean or median. Significance: *p ≤ 0.05, **p ≤ 0.01, ***p ≤ 0.001, ****p ≤ 0.0001.
(TIF)

**S1 Table. Statistics and p-values belonging to the data presented in manuscript.**
(XLSX)

**S1 File.**
(XLSX)

## Acknowledgments

We thank the Genome Engineering Zebrafish National Facility (Uppsala, Sweden) for fish husbandry.

## Author Contributions

**Conceptualization:** Remy Manuel.

**Data curation:** Melek Umay Tuz-Sasik.

**Formal analysis:** Melek Umay Tuz-Sasik.

**Funding acquisition:** Henrik Boije, Remy Manuel.

**Methodology:** Henrik Boije, Remy Manuel.

**Project administration:** Remy Manuel.

**Resources:** Henrik Boije.

**Supervision:** Henrik Boije, Remy Manuel.

**Visualization:** Melek Umay Tuz-Sasik.

**Writing – original draft:** Melek Umay Tuz-Sasik.

**Writing – review & editing:** Henrik Boije, Remy Manuel.

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
