## [Decision Letter · Decision Letter 0]

2 Mar 2022

PONE-D-22-03792Characterization of locomotor phenotypes in zebrafish larvae requires testing under both light and dark conditionsPLOS ONE

Dear Dr. Tuz-Sasik,

Thank you for submitting your manuscript to PLOS ONE. After careful consideration, we feel that it has merit but does not fully meet PLOS ONE’s publication criteria as it currently stands. Therefore, we invite you to submit a revised version of the manuscript that addresses the points raised during the review process. The expert reviewers have indicated minor revisions are needed.  Please address each comment from the reviewers, modify the manuscript as appropriate, and explain your response in rebuttal letter.  The reviewers' comments were positive and helpful.  Both and I feel that the manuscript will make a strong contribution to the zebrafish behavior research field.

We look forward to receiving your revised manuscript.

Kind regards,

James A. Marrs

Academic Editor

PLOS ONE

Journal Requirements:

Additional Editor Comments:

Please address all the comments from the 2 reviewers.

Reviewers' comments:

Reviewer's Responses to Questions

**Comments to the Author**

1. Is the manuscript technically sound, and do the data support the conclusions?

Reviewer #1: Yes

Reviewer #2: Yes

2. Has the statistical analysis been performed appropriately and rigorously? 

Reviewer #1: No

Reviewer #2: Yes

3. Have the authors made all data underlying the findings in their manuscript fully available?

Reviewer #1: Yes

Reviewer #2: Yes

4. Is the manuscript presented in an intelligible fashion and written in standard English?

Reviewer #1: Yes

Reviewer #2: Yes

5. Review Comments to the Author

Reviewer #1: Dear author,

Thank you for the very interesting manuscript.

It is very important to know the fish corrections for behavioral analytic.

I'll list below what I was wondering about.

1. Line 53-54 I get the message "incorrect illumination was used", what's "incorrect"? The behavior of the fish depends on the experimental conditions, and the results will vary depending on the conditions different from your own experimental conditions. Researchers never call it a "incorrect".

2.Why did you use "tap" for escape stimulation? If you look at the graph, you can see that there is no extreme escape under any conditions. We also need to explain the "tap" method in more detail.

3.Zebrafish are known to swim towards light (Wolf et al., 2017). Therefore, when evaluating the travel distance in a certain time unit, it is natural that the travel distance will be longer when it is bright. Repeated light-dark stimuli aim to conduct experiments more efficiently by measuring the distance traveled in a short time using the escape behavior of zebrafish. I was not sure about this experimental design and conclusion. I think it would be better to summarize it a little more concisely.

4. The number of zebrafish juveniles used in the experiment varies. In such cases, you should use a "scatter plot" instead of a bar chart. You guys use "prism" for statistical processing, so I think it's easy to do. Show me once, whether or not you use a "scatter plot".

Reviewer #2: The authors have characterised changes in free-swimming and escape response of larval zebrafish under different rearing protocols and light illumination. The manuscript has a sound research question, is written is a clear english and well-exposes methods and results. The discussion of results is also appropriate. The study adds an important piece of information to the literature with regards to the behavioural phenotype of larval zebrafish as a consequence of variability in illumination.

I suggest a minor revision, following the comments below:

Typos:

- in the abstract, line 22: please remove one of the 2 “only” - it’s repeated twice.

- in the abstract, line 23: I suggest rephrasing “one illumination setting” with “the neglected illumination setting” for better clarity

- in the abstract, line 27: please remove “display” - reveal is sufficient.

- Line 253: Please remove the full stop in this phrase “…..striped bar graphs). had……“

Comments:

- Please explain more in the discussion section (with more reference to the habituation literature) why the escape response under light conditions was weaker. E.g., exposure to daylight induces long-term habituation to light, reducing the escape response to light stimuli. Do the authors believe this was a sensory or central mechanism of habituation?

- The authors stated that larvae under a day/night cycle showed a weaker escape response in light conditions. However, it is now stated whether this is meant in terms of frequency/count of escape responses, or in terms of distance travelled (mm) ?

6. PLOS authors have the option to publish the peer review history of their article (what does this mean?). If published, this will include your full peer review and any attached files.

Reviewer #1: No

Reviewer #2: **Yes: **Carolina Beppi

---

## [Author Response · Author response to Decision Letter 0]

16 Mar 2022

Editorial comments: 

Response: We thank the Editor for this recall. Now manuscript and additional files should meet PLOS ONE’s style requirements.

Response: We thank the Editor. We addressed the concerns about “Funding Information” section.

Response: We thank the Editor for noticing. We reviewed the reference list; a book title was missing for reference [1] and some new references were added to list. Now it is complete and should be correct. None of papers in the reference list was retracted at the resubmission time. 

Response: We thank the Editor and have uploaded and checked Figures with PACE.

 

Reviewers' comments:

We would like to thank both reviewers for their efforts and constructive comments. We amended our manuscript according to their suggestions. Please find a detailed response below.

Reviewer #1: 

Dear author,

Thank you for the very interesting manuscript.

It is very important to know the fish corrections for behavioral analytic.

I'll list below what I was wondering about.

1. Line 53-54 I get the message "incorrect illumination was used", what's "incorrect"? The behavior of the fish depends on the experimental conditions, and the results will vary depending on the conditions different from your own experimental conditions. Researchers never call it a "incorrect".

Response: The authors agree with the Reviewer and have replaced that statement with “because one illumination condition was neglected” in line 54-55. 

2.Why did you use "tap" for escape stimulation? If you look at the graph, you can see that there is no extreme escape under any conditions. We also need to explain the "tap" method in more detail.

Response: The tapping device is an add-on property of the Daniovision observation chamber. It is used to evoke startle response (van den Bos et al., 2017; Faria et al., 2019; Han et al., 2020). To be more correct, we have changed escape responses with startle responses throughout revised manuscript.

We have already checked distance moved, maximum velocity values for whole experimental duration with 1-min time bins. The time bin graphs (data not shown in the manuscript) demonstrated that at each tapping stimulus, distance moved and maximum velocity were increased 4-5 times compared to baseline free-swimming behavior and also compared to resting time between tapping stimuli similar to data shown in van den Bos et al., 2017. Each group had response to tapping stimulus compared to their own baseline behavior but that comparisons did not include in the present work since our aim is to unravel the behaviors of larvae under different conditions including housing, illumination and neuromast ablation. 

In Materials and Methods, line 122-124, we explained the tapping mechanism and details in our experimental setup.

3.Zebrafish are known to swim towards light (Wolf et al., 2017). Therefore, when evaluating the travel distance in a certain time unit, it is natural that the travel distance will be longer when it is bright. Repeated light-dark stimuli aim to conduct experiments more efficiently by measuring the distance traveled in a short time using the escape behavior of zebrafish. I was not sure about this experimental design and conclusion. I think it would be better to summarize it a little more concisely.

Response: In our experiments, we don`t have illumination differences within the arena, entire arena is under either dark or light. 

Moreover, we did not use light as a startle response because we wanted to assess differences between these illumination settings independently. Therefore, we used the tapping device incorporated into the Daniovision chamber. We are unsure about the comment “measuring the distance traveled in a short time the escape behavior of zebrafish” because we believe this is exactly what we performed during our escape (startle) response analysis (time analyzed is 280 ms after tapping).

4. The number of zebrafish juveniles used in the experiment varies. In such cases, you should use a "scatter plot" instead of a bar chart. You guys use "prism" for statistical processing, so I think it's easy to do. Show me once, whether or not you use a "scatter plot".

Response: The authors agree and thank the Reviewer for the comment. The bar graphs were replaced with scatterplots for each figure. Also, the sentence in Material and Methods, line 149-151 were rewritten, also figure legends were adjusted accordingly.

Reviewer #2: 

The authors have characterised changes in free-swimming and escape response of larval zebrafish under different rearing protocols and light illumination. The manuscript has a sound research question, is written is a clear english and well-exposes methods and results. The discussion of results is also appropriate. The study adds an important piece of information to the literature with regards to the behavioural phenotype of larval zebrafish as a consequence of variability in illumination.

I suggest a minor revision, following the comments below:

Typos:

- in the abstract, line 22: please remove one of the 2 “only” - it’s repeated twice.

- in the abstract, line 23: I suggest rephrasing “one illumination setting” with “the neglected illumination setting” for better clarity

- in the abstract, line 27: please remove “display” - reveal is sufficient.

- Line 253: Please remove the full stop in this phrase “…..striped bar graphs). had……“

Response: We thank the Reviewer for bringing these typos to our attention. They have been corrected in the revised manuscript.

Comments:-

- Please explain more in the discussion section (with more reference to the habituation literature) why the escape response under light conditions was weaker. E.g., exposure to daylight induces long-term habituation to light, reducing the escape response to light stimuli. Do the authors believe this was a sensory or central mechanism of habituation?

Response: In the present study, light is not utilized to evoke escape (startle) response, instead we used mechanical/acoustic startle response by a tapping mechanism. The main purpose was to understand the behavior of fish under different, independent, illumination settings as to not miss any phenotype (Fig 1). The behavior of larvae tested under light conditions were different than larvae tested under dark conditions, as shown in the data represented with Fig 1. Thus we are not sure what the reviewer meant with “exposure to daylight induces long-term habituation to light, reducing the escape response to light stimuli”. Escape (startle) response to tapping stimuli under light condition was weaker compared to response to tapping stimuli when tested under dark condition, for larvae were housed under a day/night cycle prior to behavioral assessment. 

We do agree with the reviewer´s comment that additional words should have been devoted to the habituation effect and the light/dark differences observed. To this extend, we have added the habituation definition in line 335-353. Also, the discussion regarding a stronger startle response under darkness in line 365-366 (335-353) was expanded.

- The authors stated that larvae under a day/night cycle showed a weaker escape response in light conditions. However, it is now stated whether this is meant in terms of frequency/count of escape responses, or in terms of distance travelled (mm) ?

Response: In the revised manuscript we now clarify/specify the parameters that are weaker under light conditions: line 28-30.

---

## [Decision Letter · Decision Letter 1]

22 Mar 2022

Characterization of locomotor phenotypes in zebrafish larvae requires testing under both light and dark conditions

PONE-D-22-03792R1

Dear Dr. Tuz-Sasik,

We’re pleased to inform you that your manuscript has been judged scientifically suitable for publication and will be formally accepted for publication once it meets all outstanding technical requirements.

Kind regards,

James A. Marrs

Academic Editor

PLOS ONE

Additional Editor Comments (optional):

Congratulations on an interesting and significant contribution!

Reviewers' comments:

Reviewer's Responses to Questions

**Comments to the Author**

1. If the authors have adequately addressed your comments raised in a previous round of review and you feel that this manuscript is now acceptable for publication, you may indicate that here to bypass the “Comments to the Author” section, enter your conflict of interest statement in the “Confidential to Editor” section, and submit your "Accept" recommendation.

Reviewer #1: All comments have been addressed

Reviewer #2: All comments have been addressed

2. Is the manuscript technically sound, and do the data support the conclusions?

Reviewer #1: Yes

Reviewer #2: Yes

3. Has the statistical analysis been performed appropriately and rigorously? 

Reviewer #1: Yes

Reviewer #2: Yes

4. Have the authors made all data underlying the findings in their manuscript fully available?

Reviewer #1: Yes

Reviewer #2: Yes

5. Is the manuscript presented in an intelligible fashion and written in standard English?

Reviewer #1: Yes

Reviewer #2: Yes

6. Review Comments to the Author

Reviewer #1: Dear Authors, I recommend accepting your manuscript. I apologize for some parts of my review comments that were a little difficult to understand. As a researcher, it is not good to leave something that you do not understand, so I will explain it. In review comment 3, "Measuring zebrafish escape behavior in a short time", if the startle response is not so noticeable, it is more efficient to repeat short light and dark stimuli, for example, at 5-minute intervals. Tapping. It's a suggestion that you can collect data well. We are experimenting with this tapping system and light stimulation. I understand how you understood it, along with your answer to Review Comment 2.

Reviewer #2: I appreciate the work done by the authors to address the concerns. I have no further recommendations.

7. PLOS authors have the option to publish the peer review history of their article (what does this mean?). If published, this will include your full peer review and any attached files.

Reviewer #1: No

Reviewer #2: **Yes: **Carolina Beppi

---

## [Editor Report · Acceptance letter]

25 Mar 2022

PONE-D-22-03792R1 

Characterization of locomotor phenotypes in zebrafish larvae requires testing under both light and dark conditions 

Dear Dr. Tuz-Sasik:

I'm pleased to inform you that your manuscript has been deemed suitable for publication in PLOS ONE. Congratulations! Your manuscript is now with our production department. 

Kind regards, 

on behalf of

Dr. James A. Marrs 

Academic Editor

PLOS ONE